

# Scale-dependent relationships between soil organic carbon stocks, land-use types and biophysical characteristics in a tropical montane landscape

5  Marleen de Blécourt[1,2], Marife D. Corre[2], Ekananda Paudel[3], Rhett D. Harrison[4], Rainer Brumme[2], Edzo Veldkamp[2]

[1]Institute of Soil Science, CEN Center for Earth System Research and Sustainability, Universität Hamburg, 20146 Hamburg, Germany.

[2]Soil Science of Tropical and Subtropical Ecosystems, Büsgen-Institute, Georg-August-Universität Göttingen, 37077 Göttingen, Germany.

[3]Centre for Mountain Ecosystem Studies, Kunming Institute of Botany, Kunming 650201, Yunnan, China

[4]World Agroforestry Centre, East & Southern Africa Region, Lusaka, Zambia.

*Correspondence to:* Marleen de Blécourt (marleen.de.blecourt@uni-hamburg.de)



**Abstract.** Presently, the lack of data on soil organic carbon (SOC) in relation to land-use types and biophysical
characteristics prevents reliable estimates of carbon stocks in montane landscapes of mainland SE Asia. Our
study, conducted in a 10,000-hectare landscape in Xishuangbanna, SW China, aimed at assessing the spatial
variability in SOC and its relationships with land-use cover and key biophysical characteristics at multiple
spatial scales. We sampled 27 one-hectare plots including 10 plots in mature forests, 11 plots in regenerating or
highly disturbed forests, and six plots in open land including tea plantations or grasslands. We used a sampling
design with a hierarchical structure. The landscape was first classified according to land-use types. Within each
land-use type, sampling plots of 100 m x 100 m each were randomly selected, and within each plot we sampled
nine subplots. This hierarchical sampling design allowed partitioning of the overall variance in SOC, vegetation,
soil properties and topography that was accounted for by the variability among land-use types, among plots
nested within land-use types, and within plots. SOC concentrations and stocks did not differ significantly across
land-use types. The SOC stocks to a depth of 0.9 m were $177.6 \pm 19.6$ Mg C ha$^{-1}$ in tea plantations, $199.5 \pm 14.8$
Mg C ha$^{-1}$ in regenerating or highly disturbed forests, $228.6 \pm 19.7$ (SE) Mg C ha$^{-1}$ in mature forests, and $236.2
\pm 13.7$ Mg C ha$^{-1}$ in grasslands. In this montane landscape, variability within plots accounted for more than 50%
of the overall variance in SOC. The relationships between SOC, biophysical characteristics and land-use types
varied across spatial scales. Variability in SOC within plots was determined by tree basal area, litter layer
carbon stocks and slope. Variability in SOC among plots in open land was influenced by land-use type – SOC
concentrations and stocks in grasslands were higher than in tea plantations. In forests, the variability in SOC
among plots was related to elevation. The scale-dependent relationships between SOC and its controlling factors
demonstrate that studies which aim to investigate the land-use effects on SOC need an appropriate sampling
design reflecting the controlling factors of SOC so that land-use effects will not be masked by the variability
between and within sampling plots.




## 1. Introduction

Soils are the largest pool of terrestrial organic carbon, storing more carbon than the combined total of carbon stocks in the atmosphere and vegetation (Schlesinger, 1997). The carbon pools in soil and atmosphere are tightly linked to the photosynthetic activity of plants and decomposition of soil organic matter by soil fauna. The flux from soil organic carbon (SOC) to atmospheric $CO_2$ is one of the largest in the global carbon cycle and is sensitive to changes in land use (e.g., Powers et al., 2011) and climate (Amundson, 2001). Apart from the important role of SOC in the global carbon cycle, SOC is a dominant controlling factor of important soil functions such as soil fertility, soil structure, and soil water-holding capacity. SOC typically displays considerable spatial variability across landscapes, and understanding the drivers of this variability is essential for the development of management strategies that aim at enhancing soil functions and for SOC accounting purposes with relevance for policy makers. SOC assessments are of particular interest for the Clean Development Mechanism (CDM) and for Reducing Emissions from Deforestation and Degradation (REDD+) initiatives that aim to generate financial compensation for local communities if they protect and enhance ecosystem carbon stocks (UNFCCC, 2009).

Spatial variability in SOC is the result of soil-forming factors acting and interacting across various spatio-temporal scales (Trangmar et al., 1986). Soil-forming factors affecting SOC are soil parent material, topography, biota, human activity (which includes land-use cover and land management), time, and climate (Jenny, 1941). The importance of these controlling factors differs with spatial scale and environmental setting (Chaplot et al., 2010; Liu et al., 2013; Powers and Schlesinger, 2002). At the landscape scale, parent material (which often affects soil group, and clay type and content) is an important driver of SOC (e.g. de Koning et al., 2003; Schimel et al., 1994; Six et al., 2002). Within the same soil group, SOC is mainly influenced by land-use cover and management (e.g. de Blécourt et al., 2013; de Koning et al., 2003; Mekuria et al., 2009; Post and Kwon, 2000), and geomorphological characteristics such as slope and slope position (Chaplot et al., 2005; Corre et al., 2015; Pennock and Corre, 2001). Spatial patterns of SOC stocks are also greatly influenced by small-scale variability in biophysical factors that influence plant productivity and decomposition of soil organic matter (Hook et al., 1991; Stoyan et al., 2000). A comprehensive understanding of the sources of spatial variability of SOC and its key drivers at multiple scales is an important prerequisite for upscaling SOC data to larger areas.

In this study, we used a hierarchical sampling design to examine spatial variability in SOC and its relationships with land-use cover and key biophysical characteristics at multiple spatial scales in a tropical montane landscape in Xishuangbanna, SW China. For centuries the area's land-use cover has been characterized by swidden agriculture (also called slash-and-burn agriculture, or shifting cultivation) (Xu, 2006). The long history of swidden agriculture has resulted in a mosaic of secondary forests, agricultural fields, paddy rice, tea plantations, and rough grasslands (i.e. grasslands invaded with shrubs). Similar multi-use landscapes extend throughout SW China and the northern areas of Laos, Myanmar, Thailand and Vietnam (Garrity, 1993). In recent decades, large areas, formerly under swidden agriculture, have been transformed into landscapes with a more uniform land-use cover dominated by commercial crops and monoculture tree plantations (Rerkasem et al., 2009). The impact of the demise of swidden agriculture on ecosystem carbon stocks remains hard to predict, which is caused, among other factors, by limited SOC data (Fox et al., 2014). The few studies available for this region on SOC assessments at a landscape or national scale were conducted in northern Thailand (Aumtong et al., 2009; Pibumrung et al., 2008) and Laos (Chaplot et al., 2009, 2010; Phachomphon et al., 2010).



Our specific objectives were (i) to quantify the SOC stocks of the dominant land-use types, (ii) to determine the proportions of the overall variance in SOC and key biophysical characteristics across multiple spatial scales, and (iii) to assess the relationships between SOC, land-use types and biophysical characteristics. We sampled SOC stocks, vegetation- and soil properties, and recorded topographical parameters along a disturbance gradient ranging from tea plantation (strongly disturbed), rough grasslands, regenerating or highly disturbed forests, to mature forests (minimally disturbed). Our hierarchical sampling design allowed us to partition the overall variance of SOC, vegetation- and soil properties into the variability accounted for by land-use types, sampling plots nested within land-use types (plot distances ranging between 0.5 - 12 km), and subplots nested within the one-hectare sampling plots. Our data provide important information on SOC stocks for an understudied region, give insights into factors that drive SOC at multiple spatial scales, and will help to design better sampling strategies for SOC stocks.

## 2. Material and Methods

### 2.1 Study area

The studied landscape covered an area of about 10,000 hectare and was located in Mengsong township, Xishuangbanna prefecture, Yunnan province, China (21˚29'25.62"N, 100˚30'19.85"E), bordering with Myanmar. The topography is mountainous with elevations of 800-2000 m above sea level (asl). The climate is tropical monsoon and has a mean annual temperature (MAT) of 18 °C (at 1600 m asl). Mean annual precipitation (MAP) ranges from 1600-1800 mm, of which 80 % falls in the wet season lasting from May to October (Xu et al., 2009).

Land-use types in the area cover a disturbance gradient ranging from intensively managed tea plantation, rough grasslands, regenerating or highly disturbed forests, to mature forests, with minimal human influence. Forests in the area are classified as seasonal tropical montane rainforest in valleys, with transitions to seasonal evergreen broadleaf forest on hill slopes and ridges (Zhu et al., 2005). Dominant tree families in the forest are Lauraceae, Fagaceae, Pentaphylaceae, Euphorbiaceae, and Rubiacea (Paudel et al., 2015). Our sampling plots ranged in elevation from 1147 to 1867 m asl, with slopes up to 49% (Table 1). The soils at the sampling plots varied from Haplic and Ferralic Cambisols in narrow valleys, to Cambic and Ferralic Umbrisols and Umbric and Haplic Acrisols and Ferralsols at both midslope and upslope positions (IUSS Working Group WRB., 2006). Soil texture ranged from sandy clay loam to clay, soil pH (H$_2$O) from 3.2-6.2, and the effective cation exchange capacity (ECEC) in the subsurface soil ranged from 4.8-45.8 cmol$_c$ kg$^{-1}$ clay (Table 2).

### 2.2 Sampling design

We selected 27 one-hectare sampling plots of which 10 plots were in mature forests, 11 plots in regenerating or highly disturbed forests, and six plots were categorized as open land used as tea plantations or rough grasslands. In each sampling plot, we established nine circular subplots with a 10-m radius on a square grid with 50-m spacing. Plots were selected using double sampling for stratification, also known as two-phase sampling (Fleischer, 1990). In phase 1, we classified the land-use types of the 10,000-hectare landscape based on grid



points (400 points with 500-m spacing) that were placed on satellite images (SPOT5 acquired in 2009 and RapidEye acquired in 2010) of the study area. Each point was identified as mature forest, regenerating or highly disturbed forest, open land, or other. In phase 2, the study area was divided in 16 equal-area units. Within 12 units we randomly selected the sampling plots from the classified grid points. Minimum distance between the sampling plots was 500 m. The land-use classification of the selected sampling plots was verified through field

validations and interviews with local informants. Of the selected sampling plots, three sampling plots included a maximum of four out of the nine subplots, which did not belong to the original land-use classifications. To reduce noise in the dataset we removed these subplots from the dataset. The fieldwork, which included soil, litter and vegetation sampling, was done in 2010 and 2011.

We defined mature forests as forest sites dominated by trees with stem diameters more than 30 cm that

did not show signs of recent disturbances due to timber extraction or fire. Regenerating or highly disturbed forests included both younger forest sites dominated by smaller trees, and older forest sites that had been strongly disturbed due to timber extraction or recent burning. The selected open land plots included three plots in tea plantations and three plots in rough grasslands. Sampled tea plantations consisted of tea bushes planted in rows parallel to the slopes with few or no trees. One of the sampled tea plantation was terraced. Management

practices applied in the tea plantations involved weeding and the use of chemical fertilizers and pesticides. Weeded plants were typically left between or under the tea bushes. Rough grasslands were dominated by *Imperata cylindrica* (L.) Raeusch grass, some small shrubs and a few trees. These grasslands are typically used for extensive cattle grazing and are maintained by regular burning. We observed that some of our grassland plots burnt at least two times between 2010 and 2013. According to local informants, sampling plots in each

land-use type had been burnt in the past, as is inherent to the areas with a long history of swidden agriculture. Evidence of fire in the past was also observed by pieces of charcoal in the collected soil samples.

### 2.3 Soil and litter sampling

Soils were sampled down to 1.2 m at five depth intervals: 0-0.15 m, 0.15-0.3 m, 0.3-0.6 m, 0.6-0.9 m and 0.9-

1.2 m. At each of the nine subplots per plot, we collected samples for the top three depths from four systematically (2 m east, 2 m north, 2 m west and 2 m south of the subplot center) positioned points using a Dutch auger. Soil samples collected from each subplot were mixed thoroughly in the field to form one composite sample per sampling depth per subplot. Soil samples at 0.6-0.9-m and 0.9-1.2-m depth were taken in soil pits at four subplots and one subplot per sampling plot, respectively. These pits were also used to measure

soil bulk density for each sampling depth using the core method (Blake and Hartge, 1986). The bulk density measurements were corrected for gravel content (pebbles > 2 mm). The litter layer (including leaves, seeds, and twigs with a length < 0.2 m) was collected at each subplot with a 0.04-m$^2$ quadrant sampling frame. Samples of the litter layer were collected between May-August 2010. This one-time sampling of the litter layer coincided with the start of the rainy season and does not reflect seasonal or annual fluctuations in litterfall (Paudel et al.,

2015). The litter layer mainly consisted of fresh and partly decomposed plant material.



### 2.4 Tree inventory and topographical attributes

At all nine subplots (10-m radius) per plot we measured the diameter at breast height (DBH), at 1.3 m above the soil surface, of all trees with a DBH ≥ 10 cm. Within a 5-m radius of the subplot center we also measured the DBH of all trees with a DBH ≥ 2 cm. Tree basal area at each subplot was calculated as the sum of the basal area of all measured trees. Topographical data obtained for each subplot included slope, elevation, and compound topographic index (CTI). We measured the slope from the center of each subplot to a target point situated 5-m downslope of the subplot center using a clinometer. Elevation was derived from a SRTM digital elevation model with a 90-m resolution resampled to 30-m resolution. The CTI, also known as steady state wetness index, quantifies landscape positions based on slope and upstream contributing area orthogonal to flow direction (Gessler et al., 1995; Moore et al., 1993). High CTI values refer to valleys with large catchments and low CTI values denotes to ridges or steep slopes. We calculated the CTI from the 30-m SRTM digital elevation model using ArcGIS.

### 2.5 Laboratory analyses and calculations

We analyzed the soil samples for total organic carbon and nitrogen concentrations, soil pH, soil texture and ECEC. Litter layer samples were analyzed for total organic carbon and nitrogen concentrations. Prior to analyses, the soil samples were air dried (5 days) and sieved (< 2 mm). Litter layer samples were oven dried at 60 °C for 48 hours and weighed. Total organic carbon and nitrogen concentrations were analyzed by dry combustion for ground subsamples of each soil and litter sample using a CNS Elemental analyzer (Elementar Vario EL, Hanau, Germany). Since soil pH ($H_2O$) was below 6.2, we did not expect carbonates in these soils and carbonate removal was not necessary. Soil pH ($H_2O$), pH (KCl) and soil texture were measured on each sample from the 0-0.15-m, 0.15-0.3-m, and 0.9-1.2-m depth intervals, and on a pooled sample per sampling plot for the 0.6-0.9 m depth interval. Soil pH ($H_2O$) and pH (KCl) were measured in a 1:2.5 soil-to-solution ratio. Soil texture was determined using the pipette method distinguishing the fractions clay (<0.002 mm), silt (0.002-0.063 mm), and sand (0.063-2 mm). ECEC was measured on soil samples of the 0-0.15 m depth interval and on a pooled sample from each sampling plot for the 0.6-0.9 m depth interval. The soil samples were percolated with unbuffered 1 M $NH_4Cl$ and the percolates were analyzed for exchangeable cations using ICP-EAS (Spectroflame, Spectro Analytical Instruments, Kleve, Germany).

We calculated the litter layer organic carbon stocks using the carbon concentration, the mass of the litter layer and the sample frame area. The SOC stock for each sampling depth was calculated using:

$$SOC \ \text{stocks}(\text{Mg C ha}^{-1}) = \frac{\%C}{100} \times BD \ (Mg \ m^{-3}) \times \Delta D \ (m) \times 10{,}000 \ m^2 ha^{-1},$$

where BD is the bulk density and ΔD is the thickness of the sampling depth. Since the soil depth of some sampling plots did not reach down to 1.2 m, we reported both the total SOC stocks down to 0.9 m and the total SOC stocks down to 1.2 m. Total SOC stocks were calculated as the sum of the mean SOC stocks per sampling plot of the relevant sampling depths.



### 2.6 Statistical analyses

Statistical analyses were carried out using the statistical software R version 3.2.3 (R Core Team, 2015).
Statistical tests were conducted for each sampling depth separately. Prior to analyses, we tested the data for
normality (Shapiro-Wilk test) and equality of variances (Levene's test). Significant differences were accepted at
$P \le 0.05$, and differences at $P \le 0.1$ were considered as marginally significant.

Data at the subplot level (SOC concentrations and stocks, soil C:N ratio, other soil characteristics down
to 0.3 m, tree basal area, litter layer characteristics and topographical attributes) were analyzed using linear
mixed effects models (LME) with sampling plot included as random intercept, using the package nlme (Pinheiro
et al., 2012). We tested if land-use types (fixed effect term) differed in SOC, tree basal area, soil, litter and
topographical attributes (response variables). Multiple comparisons of the means of each land-use type were
done using Tukey's HSD test in the package multcomp (Hothorn et al., 2008). We conducted multiple
regression analyses, using LMEs with sampling plot as random intercept, to test the relationships between SOC
concentrations or stocks (response variables) with the following potential explanatory variables (fixed effect
terms): land-use type, silt-plus-clay percentage, ECEC of the subsurface soil (0.6-0.9-m depth), litter layer
carbon stock, litter layer C:N ratio, tree basal area, slope, relative elevation (change in elevation relative to the
lowest situated sampling plot), and CTI. We conducted regression analyses separately for forests (mature forest
and regenerating or highly disturbed forest combined) and open land (tea plantation and grassland combined).
Correlation tests showed that the explanatory variables included in the LMEs were not strongly correlated
(Spearmans Rho < 0.44). Minimum adequate LMEs were selected using a stepwise model selection based on the
Akaikes Information Criterion with the function stepAIC in the package MASS (Venables and Ripley, 2002).
Residuals of the selected LMEs were examined for normality and equality of variances. In cases where we
detected unequal variances, we included variance functions and if the assumption of normality was violated we
used a logarithmic transformation of the response variable. The proportion of the variance explained by the
fixed effect terms ($R^2$) of each LME was calculated according to Nakagawa and Schielzeth (2013). We used a
variance component analysis to partition the overall variance of each response variable into the variability
among land-use types, among sampling plots within land-use types, and among subplot plots within plots. For
the variance partitioning, we refitted the LME with sampling plot nested within land-use type as random
intercept. Subsequently, we tested if both random factors were required in the LMEs by leaving out the random
effect for land-use type, and comparing the two LMEs using a likelihood ratio test (Crawley, 2007).

For data that were only available at plot level (soil characteristics below 0.3-m depth other than SOC,
and SOC of the 0.9-1.2-m depth, 0-0.9-m depth and 0-1.2-m depth), we tested the effect of land-use type using
either one-way analysis of variance (ANOVA) (parametric test) followed by Tukey's HSD test, or Kruskal-
Wallis ANOVA (non-parametric test) followed by a pairwise Wilcoxon test with Holm's correction for multiple
comparisons.





## 3. Results

### 3.1 Soil, vegetation, and topographic characteristics

Comparison of soil characteristics across land-use types revealed differences in soil pH and ECEC (Table 2).
The soil pH ($H_2O$) down to 0.3 m was lowest in mature forest (data 0.15-0.3 m not shown), and the pH (KCl) down to 0.15 m was lower in mature forests than in the tea plantations. Compared to grasslands, the ECEC of the top 0.15 m was lower in tea plantations. Tree basal area and litter layer carbon stocks were higher in regenerating or highly disturbed forests and mature forests than in tea plantations and grasslands (Table 1). Litter layer C:N ratios were lower in mature forest compared to grassland. Comparison of topographical
attributes showed that the land-use types were located on similar altitudes and topographical positions (reflected by CTI). However, the tea plantations had more gentle slopes compared to the other land-use types (Table 1).

### 3.2 Soil organic carbon concentrations and stocks

We did not detect differences in SOC concentration and stocks among land-use types for any of the sampling
depths nor for the total SOC stock down to 0.9 m and 1.2 m (Figure 1, Table 3). In forests, SOC concentrations and total SOC stocks were positively associated with litter layer carbon stock, tree basal area, and elevation ($R^2$ = 0.51 for 0-0.15 m, $R^2$ = 0.25 for 0.15-0.3 m, $R^2$ = 0.18 for 0-0.9 m) (Table 4). However, the effect of elevation on total SOC stocks was only marginally significant, and for the 0.15-0.3-m depth litter layer carbon stock was the only controlling factor of SOC that was statistically significant. The effect of silt-plus-clay percentage on SOC was included in the regression LME for the 0.15-0.3-m depth but was not statistically significant (Table 4).
In open land, the most important controls of SOC concentrations and total SOC stocks were land-use type, vegetation characteristics (litter layer carbon stocks, litter layer C:N ratio and tree basal area) and slope ($R^2$ = 0.57 for 0-0.15 m, $R^2$ = 0.54 for 0.15-0.3 m, $R^2$ = 0.60 for 0-0.9 m) (Table 4). SOC concentrations and total SOC stocks increased with increasing litter layer carbon stocks and decreased with increasing slope. Furthermore,
SOC concentrations in grasslands were higher than tea plantations when controlling for the variability related to other explanatory variables (Table 4). Litter C:N ratio was included as explanatory variable for SOC at 0.15-0.3-m depth; however, this effect was marginally significant (Table 4). Tree basal area was included as explanatory factor for SOC concentrations in open land at 0.15-0.3 m and for total SOC stock, but its effects on SOC concentrations was marginally significant at 0.15-0.3 m and not significant on total SOC stocks (Table 4).


### 3.3 Variance partitioning of soil, vegetation, litter and topographical attributes

Variance partitioning showed that in the top 0.3 m of the soil, with the exception of soil pH $H_2O$ (P=0.02), land-use type did not contribute significant in soil characteristics (Figure 2a; for 0.15-0.3 m, data not shown). Variability among plots and among subplots within plots largely constituted the overall variance for SOC
concentration, total SOC stocks down to 0.9 m and soil characteristics other than soil texture (Figure 2a). For soil texture, the variability among plots was the most important component of the overall variance. Most of the overall variance in the litter layer carbon stocks and litter layer C:N ratio was accounted for by the variability among subplots within plots (Figure 2b). For tree basal area, the variability among land-use types was the most



important component of the overall variance followed by the variability within plots. The main proportion of the
overall variance in slopes was covered by the variability among subplots within plots, and the overall variance
in elevation was almost completely due the variability among plots (Figure 2b).

## 4. Discussion

### 4.1 Effects of land-use type on soil organic carbon concentrations and stocks

Our values of SOC stocks in mature forest, regenerating or highly disturbed forests, tea plantations and
grasslands (Table 3) were at the high end of the range of SOC stocks reported for these land-use types in other
studies from montane areas of mainland SE Asia (Table 5, our comparisons are based on equivalent sampling
depths). SOC stocks to a depth of 0.3 m in mature forest and regenerating or highly disturbed forest were
comparable to national estimates of SOC stocks in forests in Laos (Chaplot et al., 2010). However, our total
SOC stocks within 0-0.9-m and 0-1.2-m depth were higher than the regional estimates of SOC stocks within 1-
m depth in subtropical forests in China (Yu et al., 2011), and then the SOC stocks within the same depths in
other tropical forests in Xishuangbanna, SW China, (de Blécourt et al., 2013; Lü et al., 2010) and northern
Thailand (Aumtong et al., 2009; Pibumrung et al., 2008). Data on SOC stocks in tea plantations and grasslands
in the montane regions of SE Asia are scarce. Our observed SOC stocks in tea plantations within 0-0.6 m were
in the range of the regional estimates of SOC stocks in tea plantations reported for SW China (Li et al., 2011).
However, our values of SOC stocks within 0-1.2-m depth in grasslands were higher than the amounts reported
for fallow fields with a vegetation consisting of grasses and shrubs in northern Thailand (Aumtong et al., 2009).

Although land-use type is often considered an important controlling factor of SOC, we did not observe
differences in SOC concentrations and stocks among land-use types (Table 3, Figure 1). There are several
possible explanations for the high SOC levels in grasslands, which were similar to SOC levels in mature forests.
First, *Imperata* grasslands may have a higher belowground net primary production (NPP) compared to forests,
resulting in greater inputs of organic matter to the soil. To our knowledge, no comparable data (i.e. from sites
with similar biophysical characteristics) exists on belowground NPP in these land-use types. Belowground NPP
of regularly burnt *Imperata* grasslands in northeast India ranges from 973.8 to 1326.7 g m$^{-2}$ y$^{-1}$ (Astapati and
Das, 2010) and is far greater than the reported 111 and 379 g m$^{-2}$ y$^{-1}$ for tropical forests on Ultisols and Oxisols,
respectively (Vogt et al., 1996). Second, charcoal input in grasslands is probably relatively high due to the high
fire frequencies. However, results from field measurements on impacts of fire and charcoal additions on SOC
quantities are contradicting, ranging from SOC losses (Bird et al., 2000; Fynn et al., 2003) to no change or
increases in SOC (Eckmeier et al., 2007; Ojima et al., 1994). Studies conducted in Kalimantan, Indonesia (van
der Kamp et al., 2009; Yonekura et al., 2010) reported higher SOC stocks in *Imperata* grasslands compared to
primary forests. Similarly, a meta-study of tropical land-use conversions (Powers et al., 2011) reported an
increase in SOC stocks of 26% following forest-to-grassland conversions on soils with low activity clays and
annual precipitation of 1501-2500 mm, which are similar to the biophysical conditions in our study area.
However, this meta-study also included managed grasslands as opposed to the semi-managed (mainly by regular
burning) grasslands in our study.





The large proportion of variability within and among plots from the overall variance in SOC (Figure 2a) reflects our probability sampling technique (double sampling for stratification) for selecting plot locations. Studies with sampling designs based on prior knowledge of factors controlling SOC at a specific spatial scale of investigation (e.g. using space-for-time substitution, chronosequences, or stratification based on soil groups)

generally result in smaller variability among plots nested within land-use types, as opposed to probability sampling designs. Results of the variance component analysis showed that large variability in SOC, other soil properties, vegetation characteristics, and topography within and among sampling plots (Figure 2) masked possible land-use effects on SOC in our study area.

**4.2 Effects of biophysical characteristics and vegetation on SOC**

Our findings that the majority of the overall variance in SOC was accounted by the variability within plots and a smaller proportion was accounted by the variability among plots (Figure 2a), is similar to the findings of a study in subtropical northern New South Wales, Australia (Paul et al., 2013). A large small-scale variability was also observed on a hill slope in Laos, where 85% of the variance in SOC occurred at a 20-m scale (Chaplot et al.,

2009). In contrast, in lowland landscapes of Sumatra, Indonesia, where plots of 50 m x 50 m had slopes ranging from 3-16 %, only a small proportion of the overall variance in SOC was accounted by the variability within plots (Allen et al., 2016).

The substantial small-scale variability that we detected in SOC may reflect the heterogeneity in slope and vegetation characteristics, especially tree basal area and litter layer carbon stocks, within our one-hectare

sampling plots (Table 4 and Figure 2b). We base this on the associations of SOC with tree basal area and litter layer carbon stocks in forest, and with litter layer carbon stocks and slope in open land (Table 4), in combination with the high proportion of within-plot variability of these parameters from the overall variances (Figure 2b). We attributed the variability in SOC among plots in open land to land-use effects (tea plantations versus grasslands) whereas in forests, elevation was the most important factor controlling the variability in SOC among

plots (Table 4, Figure 2b). The low $R^2$ of our SOC-models in forests, for 0.15-0.3-m and 0-0.9-m depths, indicated a large amount of unexplained variance and suggests that other controlling factors may have contributed which we did not include in our measurements. These factors could include vegetation composition and land-use history, which we tried to document but which proved difficult to categorize meaningfully.

The observed increase in SOC in forests and open land with increasing tree basal area and litter layer

carbon stocks (Table 4) was in accordance with findings from previous studies (de Blécourt et al., 2013; Powers and Schlesinger, 2002; Woollen et al., 2012) and is attributed to biomass productivity. Enhanced biomass productivity may increase SOC input through increases in litterfall and root residues. The use of tree basal area and litter layer carbon stocks as a proxy for biomass productivity is supported by positive associations between yearly litterfall and increases in tree basal area and litter layer carbon stocks, observed in a subset of our forest

plots (Table A1, Paudel et al., 2015). The decrease in SOC with increasing slope (Table 4) was most likely related to surface erosion, which is common in montane landscapes (e.g., Arrouays et al., 1995; Corre et al., 2015). The importance of erosion and sedimentation processes on the redistribution of SOC was shown in studies conducted in Laos (Chaplot et al., 2005) and Ecuador (Corre et al., 2015); soil erosion was highest at the upper slopes and most of the eroded soil and SOC was deposited within a short distance at the lower slopes. The



observed increase in SOC in forests with increase in elevation is consistent with other studies (Chaplot et al., 2010; Dieleman et al., 2013; Powers and Schlesinger, 2002). Elevation effects on SOC are often related to changes in precipitation, temperature, soil characteristics, and biomass productivity. However, despite the large elevation gradient of the forest plots in our study (1147-1867 m asl) we did not observe any elevation effects on silt-plus-clay percentage, ECEC of the subsurface soil (reflecting clay mineralogy), soil pH $H_2O$, soil C:N ratio

or tree basal area (data not shown). Although microclimatic data for our plots were not available, the commonly occurring reduction in temperatures with increase in elevation may influence SOC decomposition rates, which could possibly explain the positive trend between elevation and SOC in our forest plots. Soil texture within a similar soil group is regarded as an important control for plant productivity, decomposition of soil organic matter, and SOC stability (Silver et al., 2000). In our study area, silt-plus-clay percentage did not influence SOC

concentrations and stocks (Table 4). Possibly the influence of soil texture on SOC was masked by the large variability in soil groups (and thus clay mineralogy) in our study area.

### 4.3 Implications for sampling soil organic carbon stocks

Probability sampling techniques as applied in our study are appropriate for assessing spatial variability of SOC

and its driving factors across scales (subplot to plot and landscape scale) but fall short in detecting land-use effects on SOC. In montane landscapes, large variability in SOC due to variability in vegetation characteristics, slope and elevation within and among plots (Figure 2) may conceal the land-use effects on SOC, unless sample sizes are very large. An often used approach that has proven to be effective in detecting land-use effects on SOC is space-for-time substitution (e.g., de Blécourt et al., 2013; de Koning et al., 2003; van Straaten et al., 2015;

Veldkamp, 1994). This approach aims to select plots that mainly differ in land-use type, with soil group and thus clay type and content, and topographical and climatic characteristics being comparable. However, in contrast to our probability sampling technique, plot selection using the space-for-time substitution approach is non random in order to meet the criteria for comparison, and thus SOC stocks measured in those studies can only be extrapolated to larger scales under similar soil group and biophysical characteristics.


### 5. Conclusions

We show that, in this tropical montane landscape in SW China, spatial variability in SOC was largest at the plot scale of one hectare. The relationships between SOC, biophysical characteristics and land-use types varied across spatial scales. The high within-plot variability in SOC reflected variability in slope and in vegetation

characteristics (litter layer and tree basal area). SOC variability among plots in forests was related to elevation, and to land-use type in open land. These scale-dependent relationships between SOC and controlling factors demonstrate that studies which investigate land-use effects on SOC require an appropriate sampling design, based on controlling factors at the scale of interest, to elucidate these effects against the background variability within and between plots.



**Author contributions**


Conceived and designed the study: Marleen de Blécourt, Rhett Harrison and Rainer Brumme. Performed the study: Marleen de Blécourt and Ekananda Paudel. Analyzed the data: Marleen de Blécourt, Marife D. Corre and Edzo Veldkamp. Wrote the paper: Marleen de Blécourt. All co-authors read and contributed to revisions of the manuscript.


**Acknowledgements**

This study was part of the project "Making the Mekong Connected - MMC" funded by the German Agency for International Cooperation (GIZ) within the German Ministry for Economic Cooperation (BMZ) (Project No. 08.7860.3- 001.00). We thank the field staff from Mengsong for their assistance. We are grateful for the

valuable support of Xu Jianchu and the staff of the World Agroforestry Centre (ICRAF, East Asia office, Kunming) in China.



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





**Table 1: Means (± SE)[a] of litter layer characteristics, tree basal area and topographical attributes of different land-use types in a tropical montane landscape in SW China.**

| Characteristics | Mature forest (n=10) | Regenerating or highly disturbed forest (n=11) | Grassland (n=3) | Tea plantation (n=3) | P value |
|---|---|---|---|---|---|
| Litter layer C concentration (%) | 40.0 (1.1) | 40.1 (1.1) | 42.8 (0.2) | 39.7 (2.3) | 0.38 |
| Litter layer C:N ratio | 29.7 (1.5) b | 36.4 (2.1) ab | 43.2 (6) a | 35 (3.4) ab | 0.02 |
| Litter layer carbon stock (Mg C ha$^{-1}$) | 5.6 (0.6) a | 4.2 (0.5) a | 1.7 (0.2) b | 1.5 (0.2) b | <0.01 |
| Tree basal area (m$^2$ ha$^{-1}$) | 29 (2.5) a | 18.2 (1.9) b | 3 (0.7) c | 0.8 (0.2) c | <0.01 |
| Slope (%) | 29.7 (1.6) a | 26.7 (1.1) ab | 31 (3.8) a | 12.9 (1.3) b | 0.05 |
| Elevation (m) | 1664 (66) | 1559 (67) | 1719 (59) | 1573 (119) | 0.54 |
| Compound topographic index[b] | 9.9 (0.4) | 8.9 (0.2) | 8.4 (0.2) | 9.8 (0.8) | 0.29 |

[a]Within a row, means followed by different letters indicate significant differences among land-use types, and means without letters indicate no significant difference among land-use types (linear mixed effects model, one-way ANOVA or Kruskal-Wallis ANOVA at P ≤ 0.05).

[b]Compound topographic index (Gessler et al., 1995; Moore et al., 1993) quantifies landscape positions based on slope and upstream contributing area orthogonal to flow direction. High CTI values refer to valleys with large

catchments and low CTI values denotes to ridges or steep slopes.



**Table 2: Means (± SE)ᵃ of soil characteristics of different land-use types in a tropical montane landscape in SW China.**

| Characteristic | Depth (m) | Mature forest (n=10) | Regenerating or highly disturbed forest (n=11) | Grassland (n=3) | Tea plantation (n=3) | P value |
|---|---|---|---|---|---|---|
| Sand (%) | 0-0.15 | 39.8 (3.9) | 36.3 (3.1) | 47.4 (3.7) | 37.6 (10.6) | 0.55 |
| | 0.6-0.9 | 40.9 (4.4) | 31.5 (4.3) | 47.5 (3.9) | 33.6 (9.3) | 0.24 |
| Silt plus clay (%) | 0-0.15 | 60.2 (3.9) | 63.7 (3.1) | 52.6 (3.7) | 62.4 (10.7) | 0.54 |
| | 0.6-0.9 | 59.1 (4.4) | 68.5 (4.3) | 52.5 (3.9) | 66.4 (9.3) | 0.24 |
| Bulk density (g cm⁻³) | 0-0.15 | 0.8 (0.05) | 0.8 (0.02) | 0.8 (0.03) | 0.7 (0.1) | 0.59 |
| | 0.6-0.9 | 1.1 (0.05) | 1.1 (0.03) | 1.0 (0.03) | 1.1 (0.0) | 0.5 |
| Soil C:N ratio | 0-0.15 | 15.1 (0.6) | 14.3 (0.4) | 16.3 (1.1) | 14.2 (0.8) | 0.21 |
| | 0.6-0.9 | 10.7 (0.5) | 10.4 (0.3) | 12.5 (0.9) | 10.4 (0.7) | 0.18 |
| pH (H$_2$O) | 0-0.15 | 4.5 (0.1) b | 4.8 (0.1) a | 5.0 (0.2) a | 5.0 (0.1) a | <0.01 |
| | 0.6-0.9 | 5.0 (0.1) | 5.0 (0.1) | 5.0 (0.2) | 4.9 (0.3) | 0.82 |
| pH (KCl) | 0-0.15 | 3.6 (0.1) b | 3.8 (0.1) ab | 3.9 (0.1) ab | 4.1 (0.1) a | 0.02 |
| | 0.6-0.9 | 3.8 (0.1) | 3.9 (0.1) | 3.9 (0.2) | 4.1 (0.1) | 0.30 |
| ECECᵇ (cmol$_c$ kg⁻¹ clay) | 0-0.15 | 47.3 (7.6) ab | 32.5 (3.6) ab | 53.6 (4.4) a | 24.1 (3.3) b | 0.04 |
| | 0.6-0.9 | 23.6 (4.3) | 16.2 (3.2) | 17.5 (1.6) | 7.7 (1.5) | 0.10 |
| Al saturation (%) | 0-0.15 | 72.4 (3.1) | 64.2 (6.1) | 60.5 (12.2) | 49.3 (12.4) | 0.27 |
| | 0.6-0.9 | 86.3 (1.4) | 80.5 (6.1) | 87.8 (1.4) | 62.8 (14.8) | 0.22 |
| Base saturation (%) | 0-0.15 | 20.5 (3.1) | 29.3 (6.0) | 35.6 (11.8) | 43.5 (11.3) | 0.12 |
| | 0.6-0.9 | 8.5 (1.5) | 12.0 (5.3) | 7.9 (1.5) | 29.1 (13.3) | 0.11 |

ᵃWithin a row, means followed by different letters indicate significant differences among land-use types, and means without letters indicate no significant difference among land-use types (linear mixed effects model, one-way ANOVA or Kruskal-Wallis ANOVA at P ≤0.05).

ᵇECEC, effective cation exchange capacity.





**Table 3: Means (± SE)[a] of soil organic carbon stocks (Mg C ha[-1]) of different land-use types in a tropical montane landscape in SW China.**

| Depth (m) | Mature forest (n = 10) | Regenerating or highly disturbed forest (n = 11) | Tea plantation (n = 3) | Grassland (n = 3) | P value |
|---|---|---|---|---|---|
| 0-0.15 | 65.5 (6.8) | 58.4 (4) | 44.3 (7) | 66 (2.6) | 0.15 |
| 0.15-0.3 | 51.7 (4.5) | 47.5 (3.7) | 40.3 (3.6) | 55.1 (5) | 0.32 |
| 0.3-0.6 | 73.4 (8) | 58.9 (4.8) | 59.1 (7.6) | 67.7 (2.6) | 0.37 |
| 0.6-0.9 | 38 (3.2) | 34.6 (3.5) | 34 (5.3) | 47.4 (4) | 0.40 |
| 0.9-1.2[b] | 18 (3.1) | 23.2 (6) | 20.8 (7.7) | 38.5 (14.9) | 0.35 |
| Sum 0-0.9 | 228.6 (19.7) | 199.5 (14.8) | 177.6 (19.6) | 236.2 (13.7) | 0.34 |
| Sum 0-1.2[b] | 252.1 (25.4) | 230.5 (24.6) | 216.2 (32.1) | 274.6 (28.2) | 0.71 |

Within a row, means followed by different letters indicate significant differences among land-use types, and
means without letters indicate no significant difference among land-use types (linear mixed effects model and one-way ANOVA at $P \leq 0.05$).

[b]The number of replicates per land-use type deviates from the original number of replicate plots because the soil depth of some sampling plots did not reach down to 1.2 m. For the 0.9-1.2-m depth and total SOC stocks to 1.2 m, the number of replication is as follows: mature forest (n = 8), regenerating or highly disturbed forest (n=8), tea plantation (n=3), grassland (n=3).



**Table 4: Coefficient estimates[a] (± SE) of effects of soil texture, vegetation characteristics and topographical attributes on SOC concentrations and total SOC stocks in forests (regenerating or highly disturbed forest and mature forest combined) and open land (tea plantation and grassland combined) in a tropical montane landscape in SW China.**

| Response | Effect | Forest (n = 21) | | Open land (n = 6) | |
|---|---|---|---|---|---|
| | | **Estimate** | **P value** | **Estimate** | **P value** |
| SOC concentration (%) at 0-0.15 m | Intercept | 2.22 (0.66) | <0.001 | 6.47 (0.48) | <0.001 |
| | Land-use type[b] | | ns | -2.01 (0.30) | <0.01 |
| | Silt-plus-clay percentage (%) | | ns | | ns |
| | ECEC[c] at 0.6-0.9 m (cmol$_c$ kg$^{-1}$ clay) | | ns | | ns |
| | Litter layer carbon stock (Mg C ha$^{-1}$) | 0.16 (0.04) | <0.001 | 0.29 (0.1) | <0.01 |
| | Litter layer C:N ratio | | ns | | ns |
| | Tree basal area (m$^2$ ha$^{-1}$) | 0.03 (0.01) | <0.001 | | ns |
| | Slope (%) | | ns | -0.04 (0.01) | <0.01 |
| | Relative elevation[d] (m) | 0.01 (0.001) | <0.01 | | ns |
| | Compound topographic Index | | ns | | ns |
| SOC concentration (%) at 0.15-0.30 m | Intercept | 0.94 (0.86) | 0.28 | 4.79 (0.54) | <0.001 |
| | Land-use type[b] | | ns | -1.64 | 0.05 |
| | Silt-plus-clay percentage (%) | 0.02 (0.01) | 0.16 | | ns |
| | ECEC[c] at 0.6-0.9 m (cmol$_c$ kg$^{-1}$ clay) | | ns | | ns |
| | Litter layer carbon stock (Mg C ha$^{-1}$) | 0.17 (0.03) | <0.001 | 0.34 (0.14) | 0.03 |
| | Litter layer C:N ratio | | ns | -0.02 (0.01) | 0.10 |
| | Tree basal area (m$^2$ ha$^{-1}$) | 0.01 (0.006) | 0.13 | -0.17 (0.08) | 0.06 |
| | Slope (%) | | ns | | ns |
| | Relative elevation[d] (m) | 0.01 (0.001) | 0.13 | | ns |
| | Compound topographic Index | | ns | | ns |
| Total SOC stock (%) at 0-0.9 m | Intercept | 109.8 (24.1) | <0.001 | 247.4 (27.1) | <0.001 |
| | Land-use type[b] | | ns | -63.22 (16.3) | 0.06 |
| | Silt-plus-clay percentage (%) | | ns | | ns |
| | ECEC[c] at 0.6-0.9 m (cmol$_c$ kg$^{-1}$ clay) | | ns | | ns |
| | Litter layer carbon stock (Mg C ha$^{-1}$) | 5.3 ( 1.53) | <0.001 | 14.27 (6.05) | 0.03 |
| | Litter layer C:N ratio | | ns | | ns |
| | Tree basal area (m$^2$ ha$^{-1}$) | 0.89 (0.35) | 0.01 | -3.53 (3.71) | 0.36 |
| | Slope (%) | | ns | -2.54 (0.87) | 0.01 |
| | Relative elevation[d] (m) | 0.08 (0.05) | 0.09 | | ns |
| | Compound topographic Index | | ns | | ns |

[a] Linear mixed effects models with sampling plot as random intercept. All effects were included in the full model, and model simplification resulted in the minimum adequate model. ns - not significant (i.e., the effects excluded by model simplifications)

[b]The land-use effect in open land is calculated as SOC in tea plantation minus SOC in grassland.



[c]ECEC, Effective Cation Exchange Capacity.

[d]Relative elevation is the change in elevation compared to the lowest situated sampling plot.





**Table 5: Overview of published soil organic carbon stocks in different land-use types from montane areas of mainland SE Asia.**

| Land use | Country, site | Soil type | Elevation (m) | Climate | | Depth (m) | SOC stock (Mg C ha⁻¹) | Reference |
|---|---|---|---|---|---|---|---|---|
| | | | | MAP (mm) | MAT (°C) | | | |
| Forest | Laos, total country | - | - | - | - | 0-0.3 | 112 | Chaplot et al. (2010) |
| | China, Xishuangbanna | Haplic Acrisol | 600 | 1539 | 21.7 | 0-1 | 84-102 | Lü et al. (2010) |
| | China, Menglong, Xishuangbanna | Ferralsols and(hyper) ferralic Cambisols | 700-830 | 1377 | 22.7 | 0-0.9 | 170 | de Blécourt et al. (2013) |
| | Thailand, Nam Hean watershed | Red Yellow Podzolic soils and Reddish Brown Lateritic soils | 215-1674 | 1405 | 16.9 (DSª)-32.5 (WSª) | 0-1 | 196.24 | Pibumrung et al. (2008) |
| | Thailand, Khun Samun Watershed | Hyperalic Alisols (Humic) and Endogleyic Luvisol (Chromic) | 300-800 | 1400 | 22-29 | 0-1.2 | ~170 | Aumtong et al. (2009) |
| | China, Subtropical zone | - | - | - | - | 0-1 | 104.4-111.2 | Yu et al. (2011) |
| Tea | China, Southwest | Haplic Acrisol | - | 1000-1700 | 15-19 | 0-0.6 | 132.3-158.7 | Li et al. (2011) |
| Fallow | Thailand, Khun Samun Watershed | Hyperalic Alisols (Humic) and Endogleyic Luvisol (Chromic) | 300-800 | 1400 | 22-29 | 0-1.2 | ~210 | Aumtong et al.(2009) |

~ an approximate value, deciphered from a figure.

ª DS - dry season, WS - wet season.





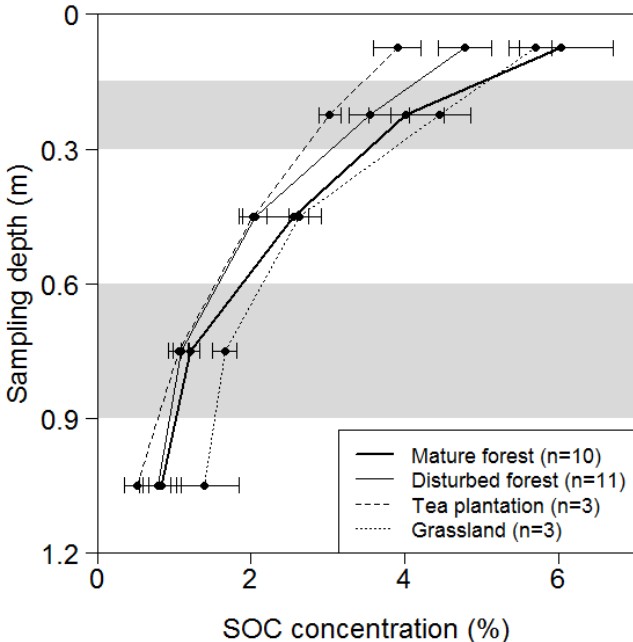

**Figure 1: Soil organic carbon concentrations in relation to sampling depth for four different land-use types in a tropical montane landscape in SW China. Alternating white and grey bands show the sampling depths. For each depth, means (SE bars) did not differ among land-use types (linear mixed effects model with P = 0.22-0.49 at sampling depths < 0.9 m, and one-way ANOVA with P = 0.37 at 0.9-1.2 m).**




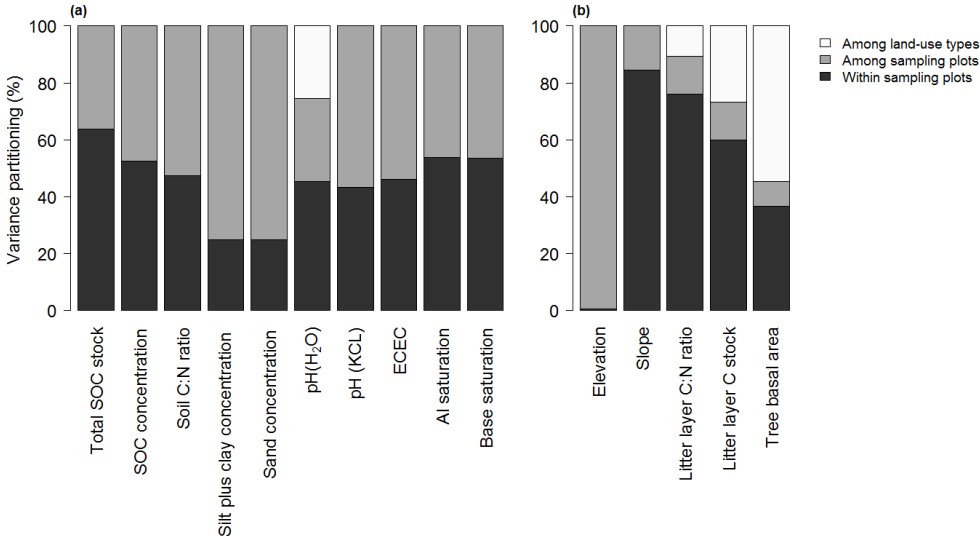


**Figure 2: Partitioning of the overall variance in (a) SOC and soil characteristics at 0-0.15-m depth, and in (b) biophysical characteristics, which can be attributed to the variability among land-use types, sampling plots nested within land-use types, and subplots nested within sampling plots (Linear mixed effects model with likelihood ratio test at P ≤ 0.05, the variability in litter C:N ratio among land-use types is marginally significant with P = 0.06)**





**Table A1. Direction of effects[a] of soil, vegetation, and topographic characteristics on litter layer carbon stock and yearly litterfall[b] in forest in a tropical montane landscape in SW China.**

| Response | Effect | Direction of effect | P value | $R^2$ |
|---|---|---|---|---|
| Litter layer carbon stock | Elevation | + | 0.05 | 0.19 |
| | Soil pH $H_2O$ (0-0.15-m depth) | - | 0.07 | |
| | Yearly litterfall | + | 0.03 | |
| Yearly litterfall | Tree basal area | + | <0.01 | 0.19 |
| | ECEC subsoil | + | 0.14 | |

[a]Linear mixed effects model with sampling plot as random intercept. Fixed effects included in the full models were elevation, slope gradient, compound topographic index, silt-plus-clay percentage, soil pH $H_2O$, tree basal area, litter C:N ratio, yearly litterfall.

[b]Litterfall was collected every month from 9 of the 21 forest plots. Details on the materials and methods are described by Paudel et al. (2015).