# Peer review of "Spatial variability of soil organic carbon in a tropical montane landscape: Associations between soil organic carbon and land use, soil properties, vegetation and topography vary across plot to landscape scales."

_SOIL, 2016_

## Referee Comment (RC1) · Anonymous Referee #1 · 30 Nov 2016

Review comments soil-2016-66 Scale-dependent relationships between soil organic carbon stocks, land-use types and biophysical characteristics in a tropical montane landscape The current rise in atmospheric CO2 concentration and associated climate change is believed to be partly mitigated by carbon sequestration in soils, primarily because the soil carbon pool comprises the biggest soil organic carbon (SOC) pool on Earth and because it exhibits direct and very dynamic exchanges with the atmosphere through photosynthesis and organic matter decomposition. This being said, the international community is seeking to identify land use and/or land management that will provide improved carbon sequestration in soils. This is where the present study by de

Blécourt et al comes in. By conducting their study in a 10,000-hectare landscape of SW China, the authors intended to estimate the link between the variability in SOC and land-use cover and topography. From the sampling of 27 one-hectare plots the authors showed that SOC concentration and stocks did not differ significantly across the land-use types from forest to tea plantation, through grasslands. However, the SOC stocks to 0.9 m increased from 177.6  $\pm$  19.6 Mg C ha-1 in tea plantations, 199.5  $\pm$  14.8 Mg C ha-1 in regenerating or highly disturbed forests, 228.6  $\pm$  19.7 Mg C ha-1 in mature forests, and 236.2  $\pm$  13.7 Mg C ha-1 in grasslands. I found the research of great value and the manuscript well written. However, before publication can be granted I would advise the authors to: 1. Ensure the conclusions are not flawed, by recalculating SOC stocks by equivalent soil mass, which is common practice; 2. Show the sampling design (plots) on a map; 3. Better consider the impact of soil type (with discussion on its correlations with topography and land use) on SOCc and SOCs. Test of variance could for instance be added to Table 4.

Finally, the abstract structure is skewed to me with half of its length on methods. I would suggest the following to be considered: Abstract A. Topic sentence (s) on the subject (its importance) and research question(s): what is(are) the research gaps in this field of research? B. Objectives of the study C. Materials and methods used in the study D. Main results (with quantitative information, tests of significance) E. Conclusions: how these results respond to the objectives; general implications of the research

---

## Referee Comment (RC2) · Anonymous Referee #2 · 2 Dec 2016

In the manuscript "Scale dependent relationships. . ." from de Blecourt at al. a comprehensive data set on SOC stocks is presented from different land use types in a region in China and their drivers including soil related variables, land-use type, topographic variables and vegetation related variables. The aim of the study was to elucidate the underlying drivers of SOC variability at scales ranging from within plot scale (< 1 ha) to between plot scale and with the same land-use type and between land use types at landscape scale (10,000 ha). The study is written well and concise. The conclusions that can be drawn from such a study a rather limited but reflect the difficulties in predicting soil organic carbon (SOC) stocks at larger than plot scale. Some major issues

have to be solved and clarified before the manuscript is in shape for publication.

Specific issues: 1.) The title is rather unclear, the readers do not know what scale the paper refers to (spatial) and whether it's a micro scale study or a global study. Also in the abstract (l. 23) the reader need to get informed about which scales are investigated. 2.) The term "biophysical characteristics" used throughout the manuscript (e.g. l. 20) is undefined and unclear. I suggest finding a better term. For example in figure 2 three parameters refer to soil organic matter, seven to soil characteristics, three to topography and three to vegetation. Instead, you combined the first to categories in fig 2a and the last two categories in 2b. I recommend to always use these three classes of drivers and one class of target variables and not to combine them randomly. 3.) The "subplots" I did not understand. The variance analysis is conducted without the subplot scale (e.g. Fig. 2). Why? If not enough driver data are available at this scale you may have to delete the subplot aspect completely. For the moment the role of the subplots are unclear. Moreover, the different numbers of samples in different depth increments (l. 145-149) may hamper a proper analysis? If still mentioned in the abstract you should provide the size of the subplots. In l. 217 you even write about "subplot plots"- what's this? 4.) You should never use SOC without specifying if you talk about SOC stocks or SOC content (e.g. l. 33). 5.) It seems to be a contradiction that you state "SOC stocks did not differ among land use types" (l. 29) but "variability of SOC (stocks?) was influenced by land use type" (l. 35). Please rephrase. 6.) You find different drivers for SOC stock variability among plots for different land use types due to a nested analysis of variance. Did you try the analysis without stratification by land use?(l. 207) 7.) Recommend to delete "with relevance for policy makers. . . .interest for" and write "for SOC accounting such as the Clean Development Mechanism. . ." in order to make it clearer. 8.) Please avoid the term "land-use cover" but only use "land-use" throughout the manuscript (e.g. 59). "Land cover" and "land use" are two different concepts. 9.) L. 62: Change clay type to clay mineralogy. 10.) L. 62 and throughout the manuscript: "soil group" should be replaced by "soil type" to make it easier to understand. 11.) L. 80-83: You mention several studies. For the reader they only make sense as introduction

into the topic if you also mention the results of these studies in relation to SOC stocks variability. 12.) L. 84: Second objective is rather unclear; please rewrite and take into account comment 1, 2 and 4. 13.) I recommend deleting paragraph 86-93. Its Material and Methods that are described in the next chapter anyway. 14.) L. 105-107: This sentence describing the forest vegetation should go to l. 132 were also the other land use types are described in detail. 15.) The studied landscape was between 800 and 2000 m asl. The plots were only between 1147 and 1867 m asl. Why did you exclude the valleys? 16.) L. 117 and l.123-126: the sampling design is difficult to follow. I recommend a figure with the sampling scheme. What are the 12 units (l. 122)? How do they refer to 16 equal area units? 17.) L. 140: The fire aspect is interesting. Was the mature forest also burned? Was there a difference in burning frequency between land use types? (l. 291). To which soil depth did you detect char coal pieces (l. 141)? 18.) L. 147: Provide the diameter of the auger. 19.) L. 171: Write the full word for ECEC the first time it appears. 20.) L. 186: The equation is wrong, since it does not take into account the stones. Stones are almost C-free and thus need to be subtracted. 21.) L. 197: Specify to which soil characteristics you are referring to. 22.) L. 205: Why was silt and clay analysed separately but taken into account for the statistical analysis only as silt-plus-clay? 23.) L. 205: Why was ECEC from topsoil no explanatory variable for SOC stocks but only subsoil ECEC? 24.) L. 209: correlated "with each other"? 25.) L. 229 and l. 239: Change "differences" to "significant differences". 26.) L. 234: Change "lower" to "narrower" 27.) L.241 and 247: To what does the R2 refers to? To the model efficiency of the regression model? If yes, you may need to rewrite this or use EF as model efficiency or the explained variance as indicator for the model performance. 28.) L. 249: Was SOC content decreasing with increasing slope for all land use types? Thus, was erosion similar among land use types (l. 336)? 29.) L. 258: Please rewrite this sentence. It is unclear. 30.) L. 270-282 and 315-317: Several other studies are mentioned here. You should also add and discuss why some other studies found other results than you. 31.) L. 294-296: Provide an explanation for this reported finding. 32.) L. 371-374 and 37-40: The conclusions are rather weak – please rewrite them.

It is nothing new that requires this additional study to find out that for the detection of land-use change effects paired plot designs are better that stratified, random or grid sampling designs. Much more interesting is where the variability of SOC stocks comes from at which scales. At which sampling plot size do we achieve representative sampling for the field site?

———————————————

---

## Author Comment (AC1) · 1 Feb 2017

We thank reviewer #1 for reviewing our manuscript. We have responded to each comment below. The line numbers refer to the lines in the original manuscript.

1. Ensure the conclusions are not flawed, by recalculating SOC stocks by equivalent soil mass, which is common practice;

Response:

We do not agree with reviewer 1 that for this study, comparisons of SOC stocks between land-use types should be based on equivalent soil mass. It is important to differentiate between studies that look at the impact of land-use changes and studies that report on spatial variability in SOC stocks and its underlying controlling factors. In the first type of studies, comparisons between the land-use types should be based on equivalent soil mass and hence at the start the study must consider the immediate reference land use prior to the present land uses being investigated. The reason for this is that land-use change often coincides with change in soil bulk density as a result of management practices, which may cause compaction or loosening of the soil. In such studies, comparison of SOC stocks due to land-use change must be based on equivalent soil mass to avoid interference of altered bulk density on SOC stock changes. However, in the second type of studies, where our present study belongs to, it is not necessary to base comparisons between land-use types on equal soil mass. Since, our objective in not about land-use change effects on SOC stocks. We aimed to quantify the spatial variability of the actual SOC stocks and its controlling factors from subplot, plot to landscape scales. Calculating SOC stocks to an equal reference mass could possibly even lead to inaccurate results as the spatial variability in soil mass contributes to the spatial variability in SOC stocks. Moreover, we have sampled the entire soil profile down to 1.2 m and no significant amounts of carbon are expected below the lowest sampling depth, meaning that the biggest impacts of any differences in soil mass between the land-use types might have been accounted for within the soil profile (VandenBygaart and Angers, 2006).

2. Show the sampling design (plots) on a map;

Response:

Reviewer 2 had also suggested that we include a map with the sampling design. We added a figure (Fig. 1) to the manuscript that will include: (1) the location of the study area in China, (2) the location of the sampling plots in the study area, and (3) a sketch of the one-ha sampling plot with the nine circular subplots arranged on a 50x50-m grid.

3. Better consider the impact of soil type (with discussion on its correlations with topography and land use) on SOCc and SOCs. Test of variance could for instance be added to Table 4.

Response:

Classification of a "soil type" is based on a range of quantitative soil properties. For instance ECEC, which is an important indicator of the presence of low or high activity clays and a key determinant parameter for soil group classification (IUSS Working Group WRB, 2014). According to the World Reference Base  (IUSS Working Group WRB, 2014), soils with low activity clays like Acrisols and Ferralsols should have a subsoil CEC $< 24$ cmolc kg$^{-1}$ clay whereas Cambisols and Umbrisol have a subsoil CEC $> 24$ cmolc kg$^{-1}$ clay. Therefore, instead of including "soil type" as a categorical factor in the linear mixed effect models we choose to test for the impact of the relevant quantitative soil properties on SOC, like silt-plus-clay percentage and ECEC of the subsoil. These soil properties have been included as explanatory factors in the full linear mixed effect models.  So, with our statistical analysis we actually considered the influence of soil type on SOC by including the quantitative soil properties "silt+clay" and ECEC and, because these are quantitative values rather than categorical variable as soil type, we are also able to quantify their influence on SOC. However, the included soil properties did not appear to have a statistically significant impact on SOC (Table 4). We also tested the impact of topography and land use on SOC, by including these factors as explanatory factors in the full linear mixed effects models.

Reviewer comment: Finally, the abstract structure is skewed to me with half of its length on methods. I would suggest the following to be considered: Abstract A. Topic sentence (s) on the subject (its importance) and research question(s): what is(are) the research gaps in this field of research? B. Objectives of the study C. Materials and methods used in the study D. Main results (with quantitative information, tests of significance) E. Conclusions: how these results respond to the objectives; general implications of the research

Response:

We agree with reviewer 1 that the quality of the abstract can be improved. We have revised the abstract by shortening the method section and following what the reviewer have outlined as the flow of the abstract. Furthermore, we provided the P values in the result's part of the abstract. We thoroughly checked that our revised abstract followed the structure described by reviewer 1.

References:
IUSS Working Group WRB: World reference base for soil resources 2014. International soil classification system for naming soils and creating legends for soil maps., 2014.

VandenBygaart, A. J. and Angers, D. A.: Towards accurate measurements of soil organic carbon stock change in agroecosystems, Can. J. Soil Sci., 86(November 2015), 465–471, doi:10.4141/S05-106, 2006.

---

## Author Comment (AC2) · 2 Feb 2017

We appreciate the thorough review and the constructive comments and suggestions by reviewer #2, the comments helped us to improve the clarity of the manuscript. Below, we address each comment. Line numbers refer to the original manuscript.

Specific issues:

1.) Reviewer comment: The title is rather unclear, the readers do not know what scale the paper refers to (spatial) and whether it's a micro scale study or a global study. Also in the abstract (l. 23) the reader need to get informed about which scales are

investigated.

Response:

We agree that the title was rather unclear. To emphasize that we look at "spatial variability" of SOC, we changed the title into: "Spatial variability of soil organic carbon in a tropical montane landscape: Associations between soil organic carbon and land use, soil properties, vegetation and topography vary across plot to landscape scales". The title now explicitly state at which spatial scales we investigated.

In the abstract at l. 23, we replaced the following sentence:"..and key biophysical characteristics at multiple spatial scales." into this sentence: ". ...and the relationships of SOC with land-use types, soil properties, vegetation characteristics and topographical attributes at three spatial scales: (1) land-use types within a landscape (10,000 ha) (2) sampling plots (one ha) nested within land-use types (plot distances ranging between 0.5 - 12 km) and (3) subplots (10-m radius) nested within sampling plots."

2.) Reviewer comment: The term "biophysical characteristics" used throughout the manuscript (e.g. l. 20) is undefined and unclear. I suggest finding a better term. For example in figure 2 three parameters refer to soil organic matter, seven to soil characteristics, three to topography and three to vegetation. Instead, you combined the first to categories in fig 2a and the last two categories in 2b. I recommend to always use these three classes of drivers and one class of target variables and not to combine them randomly.

Response:

We do understand the point of reviewer 2 that the term biophysical characteristics is vague. Therefore, when describing our own data analysis and results, instead of using the term biophysical characteristics, we now refer to: soil properties, vegetation characteristics and topographical attributes. We will change Fig 2 according to the suggestions of reviewer 2, this will result in the following three Figures: Fig 2 (a) all soil

properties, Fig 2(b) the 3 vegetation characteristics, and Fig 2 (c) the 2 topographical attributes. Regarding our target parameters, SOC stocks and SOC concentrations, we will refer to them as SOCs (for SOC stocks) and SOCc (SOC concentrations).

However, we do feel that there are multiple examples of general statements where the use of the term biophysical properties is appropriate, for example in the introduction in l. 67"…variability in biophysical factors that influence plant productivity and..", and in the discussion in l.288 "…with similar biophysical characteristics)…", and in l. 298 "….which are similar to the biophysical conditions in our study area.". In such cases, we prefer to stay with the term "biophysical characteristics".

3.) Reviewer comment: The "subplots" I did not understand. The variance analysis is conducted without the subplot scale (e.g. Fig. 2). Why? If not enough driver data are available at this scale you may have to delete the subplot aspect completely. For the moment the role of the subplots are unclear.

Response:

The variance component analysis actually did include the subplots and this is referred to in Fig. 2 as 'within sampling plots'. The "subplot" in our study is the lowest level of the sampling design – defined in statistics as the unit where the actual measurement is conducted. In our study, the subplots are single observations and contain only one data point. Our hierarchical sampling design is as follows: we sampled four land-use types in a landscape, nested within each land-use type we sampled a certain number of plots, and nested within each sampling plot we sampled nine subplots (see l. 144-145, and l. 157-161). Hence, in this study the variance component analysis includes three components: (1) variability between land-use types, (2) variability between sampling plots nested within land-use types, and (3) the variability between subplots nested within sampling plots. Exactly these three components are shown in Fig 2, in which we referred to the subplot scale as "Within sampling plots" (See caption Fig 2, l. 583). In order to make Fig. 2 clearer, we changed the legend of Fig 2 into: (-) "Among

land-use types", (-) "among plots nested within land-use types", (-) "among subplots nested within plots". Most of the driver data (soil properties, vegetation characteristics and topographical attributes) are available at the subplot level, as described in the respective methods sections l.157-161 173-174, 176-177, 180-181.

Reviewer comment: Moreover, the different numbers of samples in different depth increments (l. 145-149) may hamper a proper analysis?

Response:

It is true that we have a different number of samples in different depth increments. This number of samples for different depths was decided very carefully based on our previous research works on SOC and based on the costs of labour and analysis. This varied number of samples for each depth is not a problem for the statistical analysis, as statistical tests were conducted for each sampling depth separately (see l. 194). Moreover, we used different types of statistical tests for data available at subplot level compared to data available at plot level. For data available at subplot level (0-15, 15-30, 30-60, 60-90 cm, total SOC stocks 0-90 cm), we used linear mixed effect models (l. 197-199). For data that were only available at plot level, as is the case for depth increment 90-120, we used a one-way ANOVA or a Kruskal-Wallis ANOVA (l.221).

Reviewer comment: If still mentioned in the abstract you should provide the size of the subplots.

Response: We now included the size of the subplots within the rephrased sentence in the abstract l. 23 (see also answer to comment 1).

Reviewer comment: In l. 217 you even write about "subplot plots"- whats this?

Response: "subplot plots" at l. 217 was a typo error. We corrected this to "subplots".

4.) Reviewer comment: You should never use SOC without specifying if you talk about SOC stocks or SOC content (e.g. l. 33).
Response:

When talking about SOC quantities, it is indeed very important to know whether these quantities refer to SOC concentrations or stocks. We now checked our entire manuscript to make sure that when we report our results or findings from other studies on SOC quantities we clearly state whether we refer to stocks (SOCs) or concentrations (SOCc).

However, there are multiple examples of statements that are valid for both SOC stocks as well as SOC concentrations; for example, the paragraph in the introduction at lines 78-91. In those cases, we think it is appropriate to refer to "SOC" as a general term. We specify "SOC concentrations or stocks" wherever this particular variable is appropriate.

5.) Reviewer comment: It seems to be a contradiction that you state "SOC stocks did not differ among land use types" (l. 29) but "variability of SOC (stocks?) was influenced by land use type" (l. 35). Please rephrase.

Response:

l. 29 is based on non-significant difference among land uses (mature forest, regenerating or highly disturbed forest, and open land, Table 3) in the entire landscape, whereas l.35 was only referring to the open land category (used as tea plantations or rough grasslands, Table 4). We rephrased and restructured the abstract in order to clarify this. At l.29, we now wrote: "SOC concentrations and stocks did not differ (P values range from 0.15 to 0.71) across the four land-use types. However, within the open-land category, SOC concentrations and stocks in grasslands were higher than in tea plantations (P values range from <0.01 to 0.06)."

6.) Reviewer comment: You find different drivers for SOC stock variability among plots for different land use types due to a nested analysis of variance. Did you try the analysis without stratification by land use? (l. 207)

Response:

The reason we stratified into the categories "forest" and "open land" is to elucidate the controlling factors of SOC within a less disturbed strata "forests" versus a more human-impacted strata "open land".

As suggested by reviewer 2, we conducted the analysis for all land-use types combined (Supplement 1). We tested the same possible drivers, with the only difference that "land-use type" was not included in the linear mixed effect models to avoid multicolinearity between explanatory factors, as tree basal area and litter layer carbon stock differed between the forests and the open-land category (Table 1). This analysis showed that litter layer carbon stocks, tree basal area and elevation were positively related to SOC concentrations and stocks. These drivers (i.e. tree basal area and litter layer C stock) reflected the differences between the forest and open land categories (Table 1). The R2 for these relationships were, however, low: ranging from R2=0.17 (total SOC stocks within 0.90-m depth) to R2=0.32 (SOC concentration within 0-0.15 m). We think that the results obtained from stratifying by the forest and open land category are more meaningful. Therefore, we will not include the analysis for all land-use types combined in Table 4.

7.) Reviewer comment: Recommend to delete "with relevance for policy makers. . . .interest for" and write "for SOC accounting such as the Clean Development Mechanism. . ." in order to make it clearer.

Response:

Although we did not directly follow this recommendation of reviewer 2, we rephrased these lines in order to make it easier to understand. We changed l. 51-56 to: "Understanding the drivers of this variability is essential for the development of management strategies that aim at enhancing soil functions, and for SOC accounting purposes with a relevance for policy makers. Examples of such SOC accounting purposes are the Clean Development Mechanism (CDM) and Reducing Emissions from Deforestation and Degradation (REDD+) initiatives that aim to generate financial compensation for

local communities if they protect and enhance ecosystem carbon stocks (UNFCCC, 2009). "

8.) Reviewer comment: Please avoid the term "land-use cover" but only use "land-use" throughout the manuscript (e.g. 59). "Land cover" and "land use" are two different concepts.

Response:

We agree that land use and land-use cover are two different concepts. We replaced land-use cover with land use or land-use types. However in one case (l.78, "..more uniform land-use cover dominated by commercial crops and monoculture tree plantations.."), we would prefer to stay with land-use cover, as we think that the term land-use cover is appropriate here as this sentence refers to what actually covers the land.

9.) Reviewer comment: L. 62: Change clay type to clay mineralogy.

Response:

We changed this accordingly.

10.) Reviewer comment: L. 62 and throughout the manuscript: "soil group" should be replaced by "soil type" to make it easier to understand.

Response:

In all the sentences where we mentioned soil group, we meant here the uppermost level of soil classification, based on the FAO system World Reference Base (IUSS Working Group WRB, 2014). Soil group is the correct technical term for the meaning that we conveyed in these sentences. When you say soil type, it does not indicate which specific levels (in FAO system, subgroup to sublevel of the subgroup) of the soil classification you meant.

11.) Reviewer comment: L. 80-83: You mention several studies. For the reader they only make sense as introduction into the topic if you also mention the results of these

studies in relation to SOC stocks variability.

Response:

We listed those studies to give an idea on the number of SOC assessments conducted in this region (SW China and the northern areas of Laos, Myanmar, Thailand and Vietnam), and to show that the number of studies conducted in this region is rather small. We found only three studies from this region conducted at a landscape- or larger scale that evaluated the impact of land use and biophysical factors on the spatial variation in SOC. In previous sections of the introduction (l. 68-69), we explained that such information is necessary to upscale SOC assessments to a larger area. To give a better idea of the focus of the cited studies we revised l. 80-83 to:

"There were only three studies so far that evaluated the impact of land use and various biophysical factors on the spatial variation in SOC at a landscape or larger scale in montane mainland southeast Asia; these were conducted in northern Thailand (Aum-tong et al., 2009; Pibumrung et al., 2008) and Laos (Phachomphon et al., 2010)."

The results of the cited studies are discussed in detail and compared with our results in the discussion at l. 270-281.

12.) Reviewer comment: L. 84: Second objective is rather unclear; please rewrite and take into account comment 1, 2 and 4.

Response:

We rewrote the second objective as follows: "to determine the proportions of the overall variance of SOCc and SOCs as well as soil, vegetation and topographical properties that were accounted for by land-use types within the landscape (10,000 ha), by sampling plots (one ha) nested within land-use types (plot distances ranging between 0.5 - 12 km), and by subplots (10-m radius) nested within sampling plots,"

13.) Reviewer comment: I recommend deleting paragraph 86-93. Its Material and Methods that are described in the next chapter anyway.

Response:

We changed this by deleting l. 86-90, but we retain the last sentence as this summarizes the contribution of our unique dataset in this under-studied region.

14.) Reviewer comment: L. 105-107: This sentence describing the forest vegetation should go to l. 132 were also the other land use types are described in detail.

Response:

We changed this accordingly.

15.) Reviewer comment: The studied landscape was between 800 and 2000 m asl. The plots were only between 1147 and 1867 m asl. Why did you exclude the valleys?

Response:

We did not exclude the valleys from our sampling design; however, the information on the elevation of the studied landscape was not precisely stated in the original manuscript and we have changed this to "The topography is mountainous with elevations of 1100-1900 m above sea level (asl).

16.) Reviewer comment: L. 117 and l.123-126: the sampling design is difficult to follow. I recommend a figure with the sampling scheme. What are the 12 units (l. 122)? How do they refer to 16 equal area units?

Response:

It was indeed not clearly described how the 12 units relate to the 16 equal-area units. The study area was divided in 16 equal-area units. From these 16 units, 12 were randomly selected, and within these 12 units, we randomly selected the sampling plots from the classified grid points. We added this clarification in l.143-144.

Reviewer 1 had also suggested that we include a map with the sampling scheme. Thus, we added a figure (Fig. 1 in the revised manuscript) that shows (1) the location

of the study area in China, (2) the location of the sampling plots within the study area, and (3) a sketch of the 1-ha plot showing the nine circular subplots that were selected within the 50x50-m grid.

17.) Reviewer comment: L. 140: The fire aspect is interesting. Was the mature forest also burned? Was there a difference in burning frequency between land use types? (l. 291). To which soil depth did you detect charcoal pieces (l. 141)?

Response:

All land-use types have been burnt at some point in the past, as is inherent for areas with a long history of swidden agriculture. This information was given in l. 140. We do not have quantitative information on fire frequencies. Charcoal pieces were observed in soil samples from all depths, and we added this information in l141.

18.) Reviewer comment: L. 147: Provide the diameter of the auger.

Response:

We used an Edelman auger with 4-cm diameter. We added this information in l. 146.

19.) Reviewer comment: L. 171: Write the full word for ECEC the first time it appears.

Response:

ECEC was written full the first time it appeared in line 111: "..., and the effective cation exchange capacity (ECEC) in the subsurface soil ranged...."

20.) Reviewer comment: L. 186: The equation is wrong, since it does not take into account the stones. Stones are almost C-free and thus need to be subtracted.

Response:

We corrected the soil bulk density samples for gravel content (pebbles > 2mm) (see l. 151). By doing this, we ensured that carbon stocks are not over-estimated as in cases where soils contained gravels and soil BD was not corrected for stone content.

To improve clarity, we changed the BD definition in l. 187 to 'where BD is the soil bulk density, corrected of stone content'. Since the soils at the sampling plots were stone poor (average stone content in BD samples was 2.8 volume %, and no big stones observed in the soil profiles) we did not correct for the volume of stones in the soil profile.

21.) Reviewer comment: L. 197: Specify to which soil characteristics you are referring to.

Response:

We referred to "sand, silt plus clay, bulk density, pH (H2O), pH (KCl), ECEC, Al saturation and base saturation". We added this information in the text (l. 197) in order to make this clearer.

22.) Reviewer comment: L. 205: Why was silt and clay analysed separately but taken into account for the statistical analysis only as silt-plus-clay?

Response:

In order to determine accurately the clay and silt contents in soils, which contain iron oxides like in highly weathered Ferralsol soils, the soil must be pre-treated with a dispersing reagent (e.g. sodium dithionite) to remove iron oxides. Chemical dispersion minimizes the presence of pseudo-particles (pseudo-silt) in the particle size analysis. Although organic material had been removed prior to the soil texture analysis, our soil samples were analysed in a laboratory in China for soil texture and have not been dispersed with sodium dithionite. Therefore, we could not assure that the pseudo-particles, formed via iron oxides binding, had been dispersed, which could have resulted in an overestimation of the silt- and underestimation of the clay content. Therefore, we decided to use the sum of silt and clay content.

23.) Reviewer comment: L. 205: Why was ECEC from topsoil no explanatory variable for SOC stocks but only subsoil ECEC?

Response:

We tested ECEC as an explanatory variable of SOC, as ECEC is an important indicator of the presence of low or high activity clays and a key determinant parameter for soil group classification (IUSS Working Group WRB, 2014). However, the topsoil ECEC is not a good indicator for soil group and clay mineralogy because the topsoil's ECEC is more largely influenced by soil organic matter (that contributes also to ECEC) than the deeper soil depths. Hence, in the WRB classification system, the ECEC is specified to be based from the deeper soil depths. For example, soils with low activity clays like Acrisols and Ferralsols should have a subsoil CEC < 24 cmolc kg-1 clay whereas Cambisols and Umbrisol have a subsoil CEC > 24 cmolc kg-1 clay. Thus, we decided to use the subsoil ECEC as an explanatory variable of SOC in the linear mixed effect models instead of topsoil ECEC. We did not add an extensive clarification in the manuscript about why we used the subsoil's ECEC, as we assumed this is common knowledge to those who are familiar of what ECEC-to-SOC relationship is based on mechanistically.

24.) Reviewer comment: L. 209: correlated "with each other"?

Response:

We changed this accordingly.

25.) Reviewer comment: L. 229 and l. 239: Change "differences" to "significant differences".

Response:

We changed this accordingly.

26.) Reviewer comment: L. 234: Change "lower" to "narrower"

Response:

We changed this accordingly.

27.) Reviewer comment: L.241 and 247: To what does the R2 refers to? To the model efficiency of the regression model? If yes, you may need to rewrite this or use EF as model efficiency or the explained variance as indicator for the model performance.

Response:

R2 refers to the proportion of the variance explained by the fixed effect terms of each LME (linear mixed effect model), defined by Nakagawa and Schielzeth (2013) as marginal R2, we explained this in the method section in l.214-215. To make it easier to understand, we added this information the first time R2 appears in the results section at l. 241.

28.) Reviewer comment: L. 249: Was SOC content decreasing with increasing slope for all land use types? Thus, was erosion similar among land use types (l. 336)?

Response:

We stated in l.246 that the relationships of SOC concentrations and total SOC stocks with slope were only found for the open land category (used as grasslands and tea plantations). The lines (l. 240- l. 245) previous to this also clearly stated what controls the SOC in the forest sites, and the factors were largely vegetation-related and slope did not show significant relationship.

29.) Reviewer comment: L. 258: Please rewrite this sentence. It is unclear.

Response: In order to make the message of the sentence clearer, we revised the sentences l.257-260 to: "Variance partitioning showed that in the top 0.3 m of the soil, with the exception of soil pH H2O, land-use type did not contribute significantly to any of the variation in soil characteristics (Figure 2a; for 0.15-0.3 m, data not shown). Instead, the variability among plots (nested within land-use type) and among subplots (nested within plots) contributed relatively equally to the variances in SOCc, total SOCs down to 0.9 m and all other soil characteristics (except for soil texture)."

30.) Reviewer comment: L. 270-282 and 315-317: Several other studies are mentioned

here. You should also add and discuss why some other studies found other results than you.

Response:

In l.270-282, we described that our values of SOC stocks in the studied land-use types were at the high end of the range of SOC stocks reported for these land-use types by others studies in the region. Compared to the cited studies, our study site was located at a higher elevation (1100-1900 m asl), had a relatively low MAT (18 °C) and a relatively high MAP (1600-1800 mm) (Table 5). Elevation and MAP have commonly been observed to positively affect SOCs (Amundson, 2001; Chaplot et al., 2010; Dieleman et al., 2013) while MAT is known for being negatively associated with SOCs (Amundson, 2001; Powers et al., 2011). These factors may have contributed to the large total SOCs we observed. We added this explanation in l.282.

In l.315-317, we compared our findings with other studies; in our study, the overall variance in SOCc and SOCs was accounted by the variability within plots and only a smaller proportion was accounted by the variability among plots. In the revised manuscript, we explain why Allen et al (2016) found other results compared to the findings from our study and that of Paul et al.(2013) and Chaplot et al. (2009) as follows: "Paul et al. (2013) related the high within plot variability in SOCc to the heterogeneous nature of vegetation and microclimate in their plots. Chaplot et. al. (2009) attributed the large small-scale variation in SOCs and SOCc to land use, clay content and hill-slope surface morphology. The study of Allen et al. (2016) was on well-drained areas of the landscape with gentle slopes and stratified by soil group, which may have resulted in the small within-plot variability they observed. Our study was in a montane landscape, wherein large within-plot variability in SOCc and SOCs may have been due to a large heterogeneity in vegetation characteristics and slope within the one-ha plots (and therein the possible microclimate variability)." We elaborated on these drivers in the subsequent paragraphs l.319-322, and l.329-339.

31.) Reviewer comment: L. 294-296: Provide an explanation for this reported finding.

Response:

The findings of van der Kamp et al. (2009) and Yonekura et al. (2010) that SOC stocks in Imperata grasslands were higher than in primary forests were attributed to charcoal inputs and higher root biomass in grasslands compared to forests. We added this information in l.352-353.

32.) Reviewer comment: L. 371-374 and 37-40: The conclusions are rather weak – please rewrite them. It is nothing new that requires this additional study to find out that for the detection of land-use change effects paired plot designs are better that stratified, random or grid sampling designs. Much more interesting is where the variability of SOC stocks comes from at which scales. At which sampling plot size do we achieve representative sampling for the field site?

Response:

We revised the conclusions (l.366-374) to focus on where the variability of SOC stocks comes from at which scales and its implication on achieving representative sampling in the field. The modified conclusion is:

"In this tropical montane landscape in SW China, the spatial variability in SOCc and SOCs was largest at the plot scale. This high within-plot variability in SOC reflected the variability in litter layer carbon stocks and slope in open land, and the variability in litter layer carbon stocks and tree basal area in forests. Therefore, to achieve a reliable estimate of SOCc and SOCs within plots, it is important to have a plot size that encompasses the inherent slope and vegetation variability. Furthermore, since the variability in SOCc and SOCs among plots was related to elevation in forests, and to land-use type in open land, sampling designs for similar montane landscapes should stratify by elevation and land-use types as the principal drivers of SOC at the landscape scale. These scale-dependent relationships between SOC and controlling fac-

tors demonstrate that sampling designs must consider the controlling factors at the scale of interest in order to elucidate their effects on SOC, to control for the variability within and between plots, and to detect any possible differences in SOC between land-use types."

References:

Allen, K., Corre, M. D., Kurniawan, S., Utami, S. R. and Veldkamp, E.: Spatial variability surpasses land-use change effects on soil biochemical properties of converted lowland landscapes in Sumatra, Indonesia, Geoderma, 284, 24–50, 2016.

Amundson, R.: The Carbon Budget in Soils, Annu. Rev. Earth Planet Sci., 29, 535–562, 2001.

Aumtong, S., Magid, J., Bruun, S. and de Neergaard, A.: Relating soil carbon fractions to land use in sloping uplands in northern Thailand, Agric. Ecosyst. Environ., 131(3-4), 229–239, doi:10.1016/j.agee.2009.01.013, 2009.

Chaplot, V., Podwojewski, P., Phachomphon, K. and Valentin, C.: Soil Erosion Impact on Soil Organic Carbon Spatial Variability on Steep Tropical Slopes, Soil Sci. Soc. Am. J., 73(3), 769, doi:10.2136/sssaj2008.0031, 2009.

Chaplot, V., Bouahom, B. and Valentin, C.: Soil organic carbon stocks in Laos: spatial variations and controlling factors, Glob. Chang. Biol., 16(4), 1380–1393, doi:10.1111/j.1365-2486.2009.02013.x, 2010.

Dieleman, W. I. J., Venter, M., Ramachandra, A., Krockenberger, A. K. and Bird, M. I.: Soil carbon stocks vary predictably with altitude in tropical forests: Implications for soil carbon storage, Geoderma, 204-205, 59–67, doi:10.1016/j.geoderma.2013.04.005, 2013.

IUSS Working Group WRB: World reference base for soil resources 2014. International soil classification system for naming soils and creating legends for soil maps., 2014.

van der Kamp, J., Yassir, I. and Buurman, P.: Soil carbon changes upon secondary succession in Imperata grasslands (East Kalimantan, Indonesia), Geoderma, 149(1-2), 76–83, doi:10.1016/j.geoderma.2008.11.033, 2009.

Nakagawa, S. and Schielzeth, H.: A general and simple method for obtaining R2 from generalized linear mixed-effects models, edited by R. B. O'Hara, Methods Ecol. Evol., 4(2), 133–142, doi:10.1111/j.2041-210x.2012.00261.x, 2013.

Paul, M., Catterall, C. P. and Pollard, P. C.: Effects of spatial heterogeneity and subsample pooling on the measurement of abiotic and biotic soil properties in rainforest, pasture and reforested sites, Soil Use Manag., 29(3), 457–467, doi:10.1111/sum.12055, 2013.

Phachomphon, K., Dlamini, P. and Chaplot, V.: Estimating carbon stocks at a regional level using soil information and easily accessible auxiliary variables, Geoderma, 155(3-4), 372–380, 2010.

Pibumrung, P., Gajaseni, N. and Popan, A.: Profiles of carbon stocks in forest, reforestation and agricultural land, Northern Thailand, J. For. Res., 19(1), 11–18, doi:10.1007/s11676-008-0002-y, 2008.

Powers, J. S., Corre, M. D., Twine, T. E. and Veldkamp, E.: Geographic bias of field observations of soil carbon stocks with tropical land-use changes precludes spatial extrapolation, Proc. Natl. Acad. Sci. U. S. A., 108(15), 6318–6322, 2011.

UNFCCC: United Nations Framework Convention on Climate Change. In: Copenhagen Accord. Conference of the Parties Fifteenth Session, in Copenhagen Accord. Conference of the Parties Fifteenth Session. December 7–18., Copenhagen, Denmark., 2009.

Yonekura, Y., Ohta, S., Kiyono, Y., Aksa, D., Morisada, K., Tanaka, N. and Kanzaki, M.: Changes in soil carbon stock after deforestation and subsequent establishment of "Imperata" grassland in the Asian humid tropics, Plant Soil, 329(1), 495–507, 2010.

**SOILD**

Please also note the supplement to this comment:
http://www.soil-discuss.net/soil-2016-66/soil-2016-66-AC2-supplement.pdf

—————————————————————

Interactive
comment

[Figure]

**Supplement:**

**Table: Coefficient estimates[a] (± SE) of effects of soil texture, vegetation characteristics and topographical attributes on SOC concentrations and total SOC stocks in all land-use types (regenerating or highly disturbed forest and mature forest combined, tea plantation and grassland combined) in a tropical montane landscape in SW China.**

| Response | Effect | All land-use types combined (n=27) | |
|---|---|---|---|
| | | Estimate | P value |
| SOC concentration (%) at 0-0.15 m | Intercept | 1.52 (1.02) | 0.14 |
| | Land-use type[b] | Not included | Not included |
| | Silt-plus-clay percentage (%) | 0.01 (0.01) | 0.34 |
| | ECEC[c] at 0.6-0.9 m (cmol$_c$ kg$^{-1}$ clay) | | ns |
| | Litter layer carbon stock (Mg C ha$^{-1}$) | 0.17 (0.04) | <0.01 |
| | Litter layer C:N ratio | | |
| | Tree basal area (m$^2$ ha$^{-1}$) | 0.04 (0.01) | <0.01 |
| | Slope (%) | | ns |
| | Relative elevation[d] (m) | 0.003 (0.001) | <0.01 |
| | Compound Topographic Index | | ns |
| SOC concentration (%) at 0.15-0.30 m | Intercept | 1.41 (0.80) | 0.08 |
| | Land-use type[b] | Not included | Not included |
| | Silt-plus-clay percentage (%) | 0.008 (0.01) | 0.38 |
| | ECEC[c] at 0.6-0.9 m (cmol$_c$ kg$^{-1}$ clay) | | ns |
| | Litter layer carbon stock (Mg C ha$^{-1}$) | 0.17 (0.03) | <0.01 |
| | Litter layer C:N ratio | | ns |
| | Tree basal area (m$^2$ ha$^{-1}$) | 0.009 (0.01) | <0.01 |
| | Slope (%) | | ns |
| | Relative elevation[d] (m) | 0.002 (0.001) | 0.04 |
| | Compound Topographic Index | | ns |
| Total SOC stock (Mg C ha$^{-1}$) at 0-0.9 m | Intercept | 120.61 (22.29) | <0.01 |
| | Land-use type[b] | Not included | Not included |
| | Silt-plus-clay percentage (%) | | ns |
| | ECEC[c] at 0.6-0.9 m (cmol$_c$ kg$^{-1}$ clay) | | ns |
| | Litter layer carbon stock (Mg C ha$^{-1}$) | 5.21 (1.39) | <0.01 |
| | Litter layer C:N ratio | | ns |
| | Tree basal area (m$^2$ ha$^{-1}$) | 0.79 (0.31) | 0.01 |
| | Slope (%) | | ns |
| | Relative elevation[d] (m) | 0.07 (0.04) | 0.08 |
| | Compound Topographic Index | | ns |

[a]Linear mixed effects models with sampling plot as random intercept. All effects were included in the full model, and model simplification resulted in the minimum adequate model. ns - not significant (i.e., the effects excluded by model simplifications)

[b]Land-use type has not been included as a categorical factor in the full model.

[c]ECEC, Effective Cation Exchange Capacity.

[d]Relative elevation is the change in elevation compared to the lowest situated sampling plot.